# TGF-β-3 Induces Different Effects from TGF-β-1 and -2 on Cellular Metabolism and the Spatial Properties of the Human Trabecular Meshwork Cells

**DOI:** 10.3390/ijms24044181

**Published:** 2023-02-20

**Authors:** Megumi Watanabe, Tatsuya Sato, Yuri Tsugeno, Megumi Higashide, Masato Furuhashi, Hiroshi Ohguro

**Affiliations:** 1Departments of Ophthalmology, School of Medicine, Sapporo Medical University, Sapporo 060-8556, Japan; 2Departments of Cardiovascular, Renal and Metabolic Medicine, Sapporo Medical University, Sapporo 060-8556, Japan; 3Departments of Cellular Physiology and Signal Transduction, Sapporo Medical University, Sapporo 060-8556, Japan

**Keywords:** 3D spheroid cultures, 2D planar culture, human trabecular meshwork (HTM), TGF-β, real-time cellular metabolic analysis

## Abstract

To compare the effects among three TGF-β isoforms (TGF-β-1, TGF-β-2, and TGF-β-3) on the human trabecular meshwork (HTM), two-dimensional (2D) and three-dimensional (3D) cultures of commercially available certified immortalized HTM cells were used, and the following analyses were conducted: (1) trans-endothelial electrical resistance (TEER) and FITC dextran permeability measurements (2D); (2) a real-time cellular metabolic analysis (2D); (3) analysis of the physical property of the 3D HTM spheroids; and (4) an assessment of the gene expression levels of extracellular matrix (ECM) components (2D and 3D). All three TGF-β isoforms induced a significant increase in TEER values and a relative decrease in FITC dextran permeability in the 2D-cultured HTM cells, but these effects were the most potent in the case of TGF-β-3. The findings indicated that solutions containing 10 ng/mL of TGF-β-1, 5 ng/mL of TGF-β-2, and 1 ng/mL of TGF-β-3 had nearly comparable effects on TEER measurements. However, a real-time cellular metabolic analysis of the 2D-cultured HTM cells under these concentrations revealed that TGF-3-β induced quite different effects on the metabolic phenotype, with a decreased ATP-linked respiration, increased proton leakage, and decreased glycolytic capacity compared with TGF-β-1 and TGF-β-2. In addition, the concentrations of the three TGF-β isoforms also caused diverse effects on the physical properties of 3D HTM spheroids and the mRNA expression of ECMs and their modulators, in many of which, the effects of TGF-β-3 were markedly different from TGF-β-1 and TGF-β-2. The findings presented herein suggest that these diverse efficacies among the TGF-β isoforms, especially the unique action of TGF-β-3 toward HTM, may induce different effects within the pathogenesis of glaucoma.

## 1. Introduction

Based upon evidence-based therapy for glaucomatous optic neuropathy (GON) decreasing the intraocular pressure (IOP) to reasonable levels [1,2,3,4], is extremely important for understanding the pathogenesis of GON. Under physiological conditions, IOPs are precisely maintained by a homeostatic balance between aqueous humor (AH) production and drainage through the so-called conventional trabecular meshwork (TM) and Schlemm’s canal route as well as the uveoscleral route [5]. The elevated levels of IOPs caused by an increase in the mechanical resistance of TM due to the deposition of excessive levels of extracellular matrix (ECM) proteins are basically associated with primary open-angle glaucoma (POAG), steroid-induced glaucoma (SG), and pseudoexfoliation syndrome (PXF) [6]. Previous studies demonstrated that among three different transforming growth factor-beta isoforms (TGF-β-1~β-3) known as profibrotic cytokines, TGF-β-1 and TGF-β-2 induce the formation of these excess deposits of ECM proteins in the TM [7,8]. In fact, based on several studies using animal models as well as TM cell cultures, it has been suggested that elevated levels of TGF-β-2 in AH are produced in response to treatment with glucocorticoids [9,10,11,12]. Furthermore, it was also shown that the AH concentrations of TGF-β-2 are increased in patients with POAG but are decreased in patients with PXF, while the levels of TGF-β-1 and TGF-β-3 are substantially elevated in the AH of PXF patients compared to those with other types of glaucoma [13,14,15,16]. In addition, recent studies have also indicated that TGF-β-1 plays an important role in the pathogenesis of PXF [17,18,19,20], which is linked with oxidative stress [21,22], ER stress response [23], and dysregulated retinoic acid signaling [24]. These collective observations strongly suggest that the involvement of these three TGF isoforms (TGF-β-1, TGF-β-2, and TGF-β-3) in the pathological fibrotic changes that occur within glaucomatous TM may be different. However, most studies have focused on evaluating the TGF-β-1- and TGF-β-2-induced effects toward human trabecular meshwork (HTM) cells [13,25,26], while the effects of TGF-β-3 have been insufficiently characterized.

Therefore, in the current study, to elucidate pathophysiological roles of three TGF isoforms (TGF-β-1, TGF-β-2, and TGF-β-3) toward human TM, we used our recently developed in vitro models using two-dimensional (2D) and three-dimensional (3D) cultures, using commercially available certified immortalized HTM cells, which mimic HTM monolayers and multiple sheet structures, respectively [27,28,29,30,31].

## 2. Results

To elucidate pathological contributions of the TGF-β isoforms within glaucomatous TM, the biological effects among the TGF-β-1~3 isoforms toward 2D and 3D cell cultures of HTM cells prepared as described in our recent studies [28,30,31,32] were compared. The barrier function of the 2D cultured HTM monolayers was initially studied by measuring trans-endothelial electron resistance (TEER)/FITC dextran permeability. As shown in Figure 1, the effects of TGF-β-3 included significantly increased TEER values (panel A) and relatively decreased FITC dextran permeabilities (panel B), and these effects were concentration-dependent. However, in contrast to TGF-β-3, TGF-β-1 and TGF-β-2 exhibited similar but distinctly different effects within a concentration range of 1–10 ng/mL. That is, (1) the TGF-β-1 induced an increase in the TEER values was higher at a concentration of 10 ng/mL than at the 1 ng/mL, and (2) the TGF-β-2 induced an increase in the TEER values that reached a plateau at a concentration of 5 ng/mL, but no increasing effects were detected at a concentration of 1 ng/mL. Alternatively, the FITC–dextran permeabilities fluctuated in an opposite manner to the TEER values despite the fact that the differences were not significant. Taken together, although these effects were different depending on the specific isoform, the enhancement effects with respect to TEER values were nearly comparable for 10 ng/mL solutions of TGF-β-1, 5 ng/mL solutions of TGF-β-2, and 1 ng/mL solutions of TGF-β-3 (Figure 1). This provided us with a rationale for comparing other biological functions among the three TGF-β isoforms. Therefore, additional experiments were conducted at these fixed TGF-β isoform concentrations (10 ng/mL TGF-β-1, 5 ng/mL TGF-β-2, and 1 ng/mL TGF-β-3). 

The effects of the TGF-β isoforms on cellular metabolic functions of 2D-cultured HTM cells are shown in Figure 2. Consistent with our previous reports [32], treatment with TGF-β-1 or TGF-β-2 resulted in a shift in intracellular metabolism from mitochondrial respiration to glycolysis in the case of the 2D-cultured HTM cells. However, unlike TGF-β-1 or TGF-β-2, the treatment with TGF-β-3 resulted in decreased ATP-linked respiration, increased proton leakage, and suppressed glycolysis. These results suggest that TGF-β-3 induces different metabolic effects from TGF-β-1 or TGF-β-2 in 2D-cultured HTM cells.

Next, to estimate the effects of the TGF-β isoforms toward HTM’s multiple layers of sheets [33], our recently established 3D HTM spheroid culture systems [27,28,30,31] was employed, and their physical aspects, size, and stiffness were compared among them. As shown in Figure 3, in the presence of a 10 ng/mL solution of TGF-β-1, the 3D HTM spheroids were significantly down-sized as compared with non-treated control (CONT) and TGF-β2- or -3 (panel A). Although the stiffness values were not different between the CONT and TGF-β isoforms, the TGF-β-treated 3D spheroids were substantially softened as compared with TGFβ-1- or -2-treated isoforms (panel B).

To study these issues in more detail, the effects of these TGF-β isoforms on the expression of some major ECM proteins including collagen1 (COL1), COL4, COL6, fibronectin (FN), and α smooth muscle actin (αSMA) were determined by qPCR analysis (Figure 4). In the presence of the TGF-β isoforms, the mRNA expression of COL6 of the 2D- or 3D-cultured HTM cells were significantly or relatively down-regulated, respectively, but the other ECMs were substantially up-regulated. Such TGF-β-3-induced up-regulatory effects were less than that for TGF-β-1 or -2. Although the statistical significance was not detected, the immunolabeling intensities of those ECM proteins also similarly fluctuated (Figure 5). As possible reason for the discrepancy between the stain intensities and the gene expression levels of the 3D spheroid specimens, it was speculated that the immunolabeling reflected the target molecules localized within the surface of the 3D spheroid, but in contrast, the gene expressions analysis detected the levels of the target molecules within whole 3D spheroid. In fact, such discrepancy was also observed in our previous studies [34,35]. Similar to this, dominant effects of TGF-β-1 or -2 were also observed in the mRNA expressions of some of the ECM modulators, namely TIMPs (Figure 6) and MMPs (Figure 7). That is, a significant up-regulation of TIMP 2 (3D) by TGF-β-1 or -2, TIMP3 (2D) by TGF-β-1, and MMP2 (2D and 3D) by TGF-β-1 or -2 and down-regulation of MMP9 by TGF-β-1 (2D) were observed.

## 3. Discussion

As a rationale for using the 3D HTM spheroid model in addition to the conventional 2D HTM cells in the current investigation, in our recent study [28], we found that the normally observed down-sizing effects during the 6-day culture were further enhanced by dexamethasone (DEX) and TGF-β-2 and that both drugs caused the formation of substantial ECM deposits, in which these effects were more evident by TGF-β-2. Concerning structural aspects, the 3D spheroid consists of multiple layers of HTM cells that were arranged concentrically within the 3D HTM spheroid. Based upon these results, we concluded that our established 3D HTM spheroids may more accurately replicate multiple sheet structure of the human TM structure. In addition, the different increase of ECM deposits by TGF-β-2 or DEX within the 3D HTM spheroid suggests that these may become POAG or a steroid-induced glaucoma model.

Since the cloning of the human TGF-β-1 [36], additional members of the human TGF-β family have been identified, including three TGF-β isoforms (TGF-β-1, -2, and -3), activins, nodal, bone morphogenetic proteins (BMPs), and growth and differentiation factors (GDFs) [37]. Functionally, it has been reported that the TGF-β-1, -2, and -3 isoforms induce different effects in controlling wound-healing processes despite the fact that these isoforms function through the same receptors and the Smad2/3 signaling pathway [38]. It is noteworthy that, although TGF-β-1 or -β-2 promotes ECM deposition in the early stages of wound healing, the final quality of wound scarring is essentially the same between untreated wounds or wounds that have been treated with TGF-β-1 or -β-2 [38]. However, in contrast, the inhibition of TGF-β/Smad3 signaling by the knockdown of Smad3 expression or by the presence of an antibody to TGF-β-1 and -β-2 significantly reduces scarring, thus resulting in a higher quality of wound healing [39,40]. Alternatively, since the administration of TGF-β-3 but not TGF-β-1 or -β-2 reduces cutaneous scarring [38], a recombinant TGF-β-3 ligand was tested for prophylactic use in a phase I/II trial [41]. Similar to these diverse effects among three TGF-β isoforms toward wound-healing processes, these three isoforms identified within AH may also be involved in the pathogenesis within the different types of glaucoma in different manners. That is, increases in the AH concentrations of TGF-β-2 or TGF-β-1 and TGF-β-2 were detected in patients with POAG [42,43] or primary angle-closure glaucoma (PACG) [44], respectively. Alternatively, it was also shown that the AH concentrations of TGF-β-1 were higher in eyes with the pseudoexfoliation syndrome (PXF) and pseudoexfoliation glaucoma (PXG) than controls or other primary types of glaucoma [20,45]. However, other studies have reported that all TGF-β-1~TGF-β-3 isoforms were also elevated in terms of AH levels of TGF-β isoforms in patients with PXF and PXG [15]. Therefore, these collective observations rationally suggest that the TGF-β-1~TGF-β-3 isoforms induce different effects toward HTM, which is the one of most important biological segments that regulates AH outflow among several types of glaucoma. In the current study, to elucidate the pathological effects TGF-β isoforms toward HTM, we recently developed in vitro models that replicate the structures of monolayers and multiple layers of HTM using 2D and 3D cell cultures of immortalized HTM cells. As a result, the following findings were obtained: (1) all three TGF-β isoforms increase the barrier functions of 2D HTM cell monolayers based on TEER measurements although the efficacy of TGF-β-3 was the most potent among them. In addition, regarding the specific concentrations of each isoform required to induce the same efficacies toward TEER measurements, (2) the TGF-β-3-induced distinct effects on the cellular metabolic states of the 2D-cultured HTM cells and (3) the size or stiffness of the 3D HTM spheroids were significantly decreased in the case of TGF-β-1 or TGF-β-3, respectively, as compared with the others. Thus, these characteristic biological features of TGF-β isoforms may contribute to different types of the glaucoma pathogenesis. 

In our real-time cellular metabolic measurements using a Seahorse Bioanalyzer, quite interestingly, TGF-β-3, despite being the “least” concentrated, dramatically altered cellular metabolism compared with TGF-β-1 and TGF-β-2 (Figure 2). This result strongly suggests that TGF-β-3 induces a different type of intracellular signaling from TGF-β-1 and TGF-β-2 in altering cellular metabolism. To the best of our knowledge, there have been no reports in which isoform-specific pharmacological differences in receptor signaling have been examined in HTM cells. However, a previous report by Hall et al. [46] showed that mice lacking TGF-β-1 but instead knocking in TGF-β-3 did not show embryonic lethality, as seen in TGF-β1-deficient mice, but had different phenotypes (shorter life span, tooth and bone loss, and altered white fat cellular metabolism), suggesting that there is at least some incompatibility between TGF-β-1- and TGF-β-3-induced cellular signaling. Furthermore, HTM cells that had been treated with TGF-β-3 showed decreased ATP-linked respiration and increased proton leakage with a concomitant suppression of glycolysis, suggesting that TGF-β-3 induced aberrantly enhanced uncoupled respiration. Since it is known that TGF-β causes an increase in the production of reactive oxygen species (ROS) via activated NADPH oxidases (NOXs) [47], it is possible that ROS-induced mitochondrial inner membrane damage could result in aberrantly increased uncoupled respiration in HTM cells that were treated with TGF-β-3. Another possibility is that this enhanced uncoupled respiration could have resulted from compensatory activated uncoupling proteins (UCPs) that can cancel or regulate increased ROS production [48]. In fact, pro-inflammatory cytokines such as interleukin 6 (IL-6), interleukin 8 (IL-8), tumor necrosis factor-alpha (TNF-α), and vascular endothelial growth factor (VEGF) and others have been reported to function in pro-fibrogenetic environment [49,50,51,52] under conditions of oxidative stress, which are known to be involved in the pathogenesis of PXF and PXG [52,53]. 

In the current study, we observed significant changes of the gene expressions of the ECM modulating factors TIMPs and MMPs in the presence of TGF-β isoforms. That is, TGF-β-1 induced down-regulation of MMP9 and up-regulations of MMP2, TIMP2, and -3; and TGF-β-2 and TGF-β-3 caused up-regulation of MMP2. In fact, it is recognized that ECM remodeling by MMPs stimulates AH outflow through the TM, based on a study using a perfused model of human anterior segment cultures [54]. In addition, using a MMP9-knockout mice model with ocular hypertension, De Groef et al. reported that the remodeling of the TM by MMP9 is required to enhance the outflow and maintain IOP homeostasis [55]. More interestingly, Sahay et al. reported that TGF-β-1 levels as well as MMP9 were increased in the tears in AH in patients with PXF and PXG as compared to POAG, suggesting that the elevated TGF-β-1 levels may affect ECM remodeling mechanisms [56]. Therefore, based upon these collective observations, we rationally speculate that TGF-β isoforms induce an imbalance between the MMP/TIMP ratio and that this may contribute to the pathogenesis associated with the decreased outflow and elevated IOP in glaucomatous eyes. In addition, some of the different effects by TGF-β isoforms on cellular metabolic functions as well as the physical properties of the 3D HTM spheroids, even at same efficacies toward TEER measurements, may be involved in the different types of glaucomatous eyes. However, since these concepts remain speculative at present, and currently used immortalized TM cells do not always behave the same as primary HTM cells, additional investigations using additional methodology such as RNA sequencing as well as the inhibition of specific candidate molecules by SiRNA and others in addition to using the same concentrations of TGF-β isoforms and primary HTM cells obtained from several glaucoma and non-glaucoma individuals will be required.

## 4. Materials and Methods

### 4.1. 2D and 3D Cell Cultures of Human Trabecular Meshwork (HTM) Cells

All current experiments using human tissue/cells were conducted in compliance with the tenets of the Declaration of Helsinki and were approved by the internal review board of Sapporo Medical University. Commercially available, certified immortalized HTM cells (Applied Biological Materials Inc., Richmond, BC, Canada) were used in this study. The authenticity of these HTM cells was independently verified as described in Appendix A.

Two-dimensional cultures of the HTM cells were prepared as described in a previous report [27]. Briefly, HTM cells, which were used after the 20th passage, were maintained in 150 mm 2D planer culture dishes at 37 °C in HG-DMEM medium containing 10% FBS, 1% L-glutamine, and 1% antibiotic-antimycotic until reaching 90% confluence by changing the medium every other day.

The 3D spheroids of HTM were prepared using hanging droplet spheroid three-dimension (3D) culture plates (# HDP1385, Sigma-Aldrich Co, St. Louis, MO, USA) as described in a previous report [57]. In brief, after collecting and resuspending the 2D cultured HTM cells as above, 20,000 HTM cells in 28 μL of the above medium supplemented with 0.25% methylcellulose were 3D-cultured for a period of 6 days. At day 1, the 1, 5, or 10 ng/mL solutions of TGFβ-1, -2, or -3 [27] were added, and thereafter, half of the medium in each well was exchanged on every following day.

### 4.2. TEER and FITC Dextran Permeability Measurements of 2D-Cultured HTM Monolayer

The TEER and FITC dextran permeability measurements of HTM cell monolayers were carried out according to previously described methods [58,59]. (Information concerning the methods used in this study are shown in Appendix A). In terms of the normalization for cell densities among different experimental conditions, we confirmed that the numbers of the nuclear staining by DAPI were nearly comparable among these during their immunocyte chemistry.

### 4.3. Analysis of Real-Time Cellular Metabolism of the 2D-Cultured HTM Cells by a Seahorse Bioanalyzer

The real-time cellular metabolic function analysis involved collecting data on the oxygen consumption rate (OCR) and extracellular acidification rate (ECAR) of the 2D HTM cells that had been cultured in the absence and presence of the three TGF-β isoforms for 6 days using a Seahorse XFe96 Bioanalyzer (Agilent Technologies, Santa Clara, CA, USA) as described recently [32,59,60]. (Information concerning the methods used in this study are shown in Appendix A.)

### 4.4. Immunocytochemistry of 2D-Cultured HTM Cells and 3D HTM Spheroids

The immunocytochemistry of the 2D-cultured HTM cells and 3D HTM spheroids was examined by previously described methods, with minor modifications [61,62]. (Information concerning the methods used in this study are shown in Appendix A).

### 4.5. Characterization of the Physical Properties, Sizes, and Stiffness, of the 3D HTM Spheroid

Measurements of the mean size (μm^2^) and stiffness as force/displacement (μN/μm) of individual 3D HTM spheroids were performed as described previously [57,61]. (Information concerning the methods used in this study are shown in Appendix A).

### 4.6. Other Analytical Methods

Total RNA extraction followed by reverse transcription and real-time PCR were conducted were described previously [27] using specific primers and Taqman probes shown in Appendix A. For the quantification, 36B4 was used as a reference gene. (Information concerning the methods used in this study are shown in Appendix A.)

All statistical analyses involved the use of the Graph Pad Prism 8 (GraphPad Software, San Diego, CA, USA) as described previously [27]. All data are presented as arithmetic mean ± the standard error of the mean (SEM), and statistical significance among experimental groups was evaluated by ANOVA followed by a Tukey’s multiple comparison test.

## Figures and Tables

**Figure 1 ijms-24-04181-f001:**
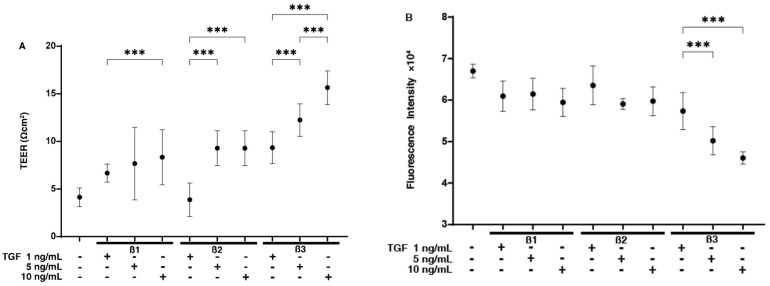
Effects of TGF-β isoforms on the barrier functions of 2D cultures of HTM cell monolayers by trans-endothelial electrical resistance (TEER) (**A**) and FITC-dextran permeability measurements (**B**). In the presence or absence of 1, 5, or 10 ng/mL TGF-β-1, TGF-β-2, or TGF-β-3, the 2D cultured HTM monolayers were subjected to TEER and FITC permeability measurements. Plots of the TEER values (Ωcm^2^) are shown in panel A, and the fluorescein intensities at an excitation wavelength of 490 and an emission wavelength of 530 nm are plotted in panel B. The numbers of cells within the 100 μm × 100 μL squares (n = 10 different area) in the TEER 2D planar culture plate (NT; 4.2 ± 1.17, TGF-β-1; 4.4 ± 1.85, TGF-β-2; 5.0 ± 1.53, TGF-β-3; 4.6 ± 0.93). All experiments were performed in triplicate using fresh preparations (n = 4). *** *p* < 0.005.

**Figure 2 ijms-24-04181-f002:**
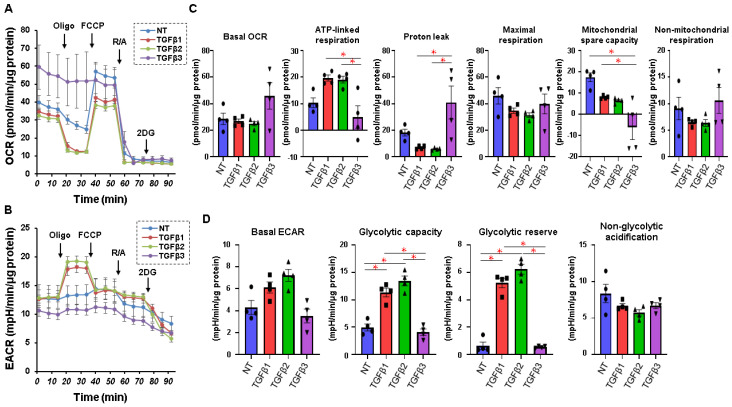
Effects of three TGF-β isoforms on cellular metabolic phenotypes in 2D-cultured HTM cells. At day 6, 2D-cultured HTM (NT, non-treated control) and those treated with 10 ng/mL solutions of TGF-β-1, 5 ng/mL solutions of TGF-β-2, or 1 ng/mL solutions of TGF-β-3 were subjected to a real-time metabolic function analysis using a Seahorse XFe96 Bioanalyzer. Panel (**A**,**B**): Simultaneous measurements of oxygen consumption rate (OCR, Panel (**A**)) and extracellular acidification rate (ECAR, Panel (**B**)) at the baseline and those by subsequent supplementation with oligomycin (Oligo, a complex V inhibitor), carbonyl cyanide-p-trifluoromethoxyphenylhydrazone (FCCP, a protonphore), rotenone/antimycin A (R/A, complex I/III inhibitors), and 2-deoxy-D-glucose (2DG, a hexokinase inhibitor). Panel (**C**,**D**): Key parameters of mitochondrial respiration (Panel (**C**)) and glycolytic flux (Panel (**D**)). Basal OCR was calculated by subtracting OCR with rotenone/antimycin A from the OCR at baseline. ATP-linked respiration was calculated by subtracting the OCR with oligomycin from the OCR at the baseline. Proton leakage was calculated by subtracting the OCR in the presence of rotenone/antimycin A from the OCR in the presence of oligomycin. Maximal respiration was calculated by subtracting the OCR in the presence of rotenone/antimycin A from the OCR in the presence of FCCP. Mitochondrial spare capacity was calculated by subtracting the OCR at the baseline from the OCR in the presence of FCCP. Non-mitochondrial respiration was defined as OCR with rotenone/antimycin A. Basal ECAR was calculated by subtracting ECAR with 2DG from ECAR at the baseline. Glycolytic capacity was calculated by subtracting ECAR with 2DG from ECAR with oligomycin. Glycolytic reserve was calculated by subtracting ECAR at baseline from ECAR with oligomycin. Non-glycolytic acidification was defined as the final value of ECAR in the presence of 2DG. All experiments were performed using fresh preparations (n = 4). * *p* < 0.05.

**Figure 3 ijms-24-04181-f003:**
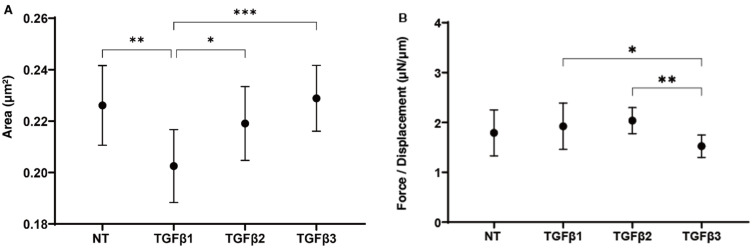
Effects of TGF-β isoforms on the physical properties, i.e., size (**A**) and stiffness (**B**), of 3D HTM spheroids. Among the non-treated control (NT) cultures and cultures treated with 10 ng/mL TGF-β-1, 5 ng/mL TGF-β-2, or 1 ng/mL TGF-β-3, the mean sizes of the 3D HTM spheroids are plotted in panel (**A**). Under these conditions, the stiffness of 3D HTM spheroids, based on the force required (μN) to induce a 50% deformity in diameter, was measured by a micro-squeezer, and force/displacement (μN/μm) values are plotted in panel (**B**). These experiments were performed in triplicate using fresh preparations (n = 10, total 30, and 15, total 45, for size measurement and stiffness analysis, respectively). * *p* < 0.05; ** *p* < 0.01; *** *p* < 0.005.

**Figure 4 ijms-24-04181-f004:**
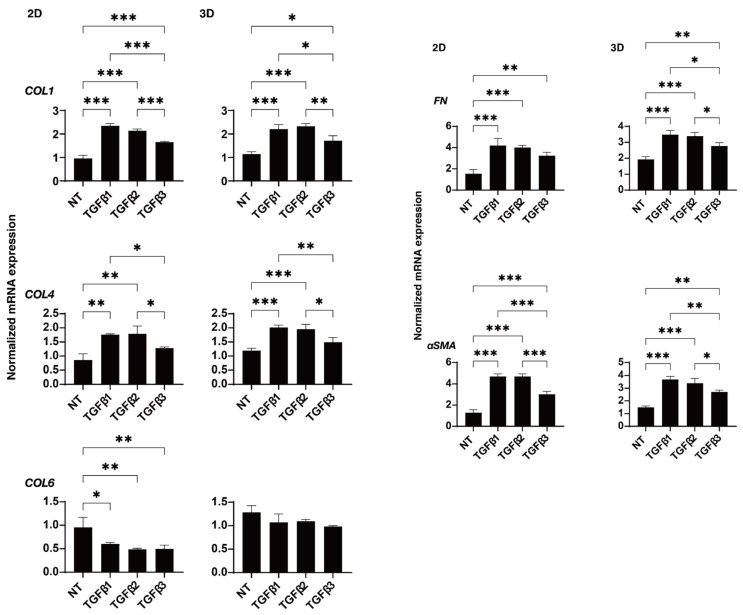
Effects of TGF-β isoforms on the mRNA expression of ECM molecules in 2D- and 3D-cultured HTM cells. At day 6, 2D- and 3D-cultured HTM cells (NT, non-treated control) and cultures treated with 10 ng/mL TGF-β-1, 5 ng/mL TGF-β-2, or 1 ng/mL TGF-β-3 were subjected to a qPCR analysis to estimate the expression of mRNA in ECMs (*COL1*, *COL4*, *COL6*, *FN,* and *aSMA*). All experiments were performed in triplicate using freshly prepared 2D HTM cells and 3D HTM spheroids (n = 15–20, total 45–60) in each experimental condition. * *p* < 0.05; ** *p* < 0.01; *** *p* < 0.005.

**Figure 5 ijms-24-04181-f005:**
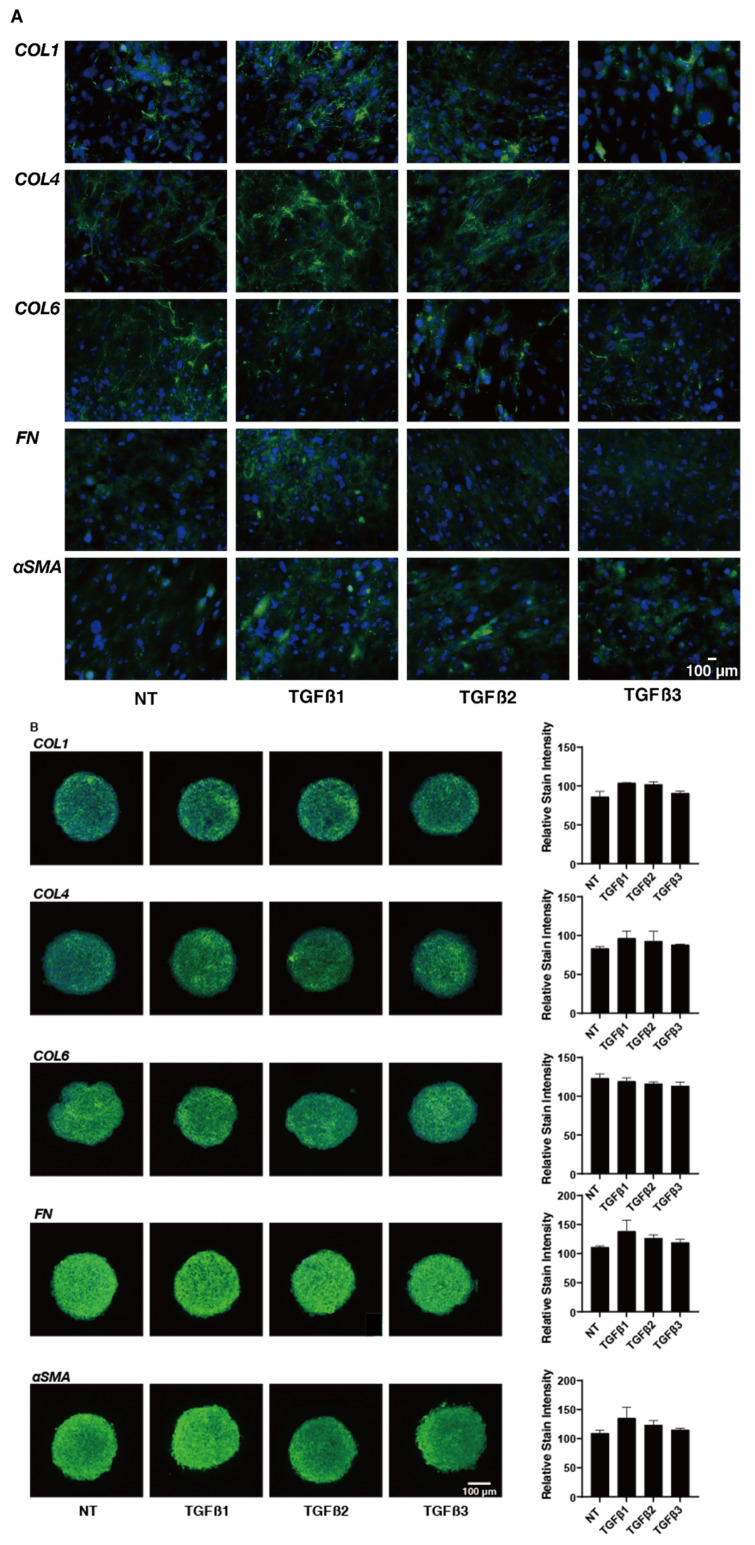
Effects of TGF-β isoforms on immunolabeling against ECM molecules in 2D- and 3D-cultured HTM cells. At day 6, 2D- and 3D-cultured HTM cells (NT, non-treated control) and cultures treated with 10 ng/mL TGF-β-1, 5 ng/mL TGF-β-2, or 1 ng/mL TGF-β-3 were subjected to immunofluorescent labeling of ECMs (*COL1*, *COL4*, *COL6*, *FN,* and *aSMA*). All experiments were performed in duplicate using fresh preparations (n = 5, total 10). Representative merged images with DAPI (blue) and ECM (green) are shown (panel (**A**) **left**; 2D, panel (**B**) **left**; 3D, scale bar; 100 μm). The immunostaining levels of each target molecule among the observed areas were evaluated using Image J (NIS-Elements 4.0 software) and plotted (panel (**A**) **right**; 2D, panel (**B**) **right**; 3D).

**Figure 6 ijms-24-04181-f006:**
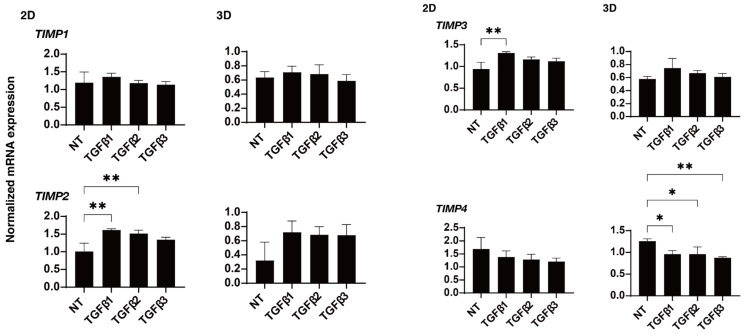
Effects of TGF-β isoforms on the mRNA expression of TIMPs in 2D- and 3D-cultured HTM cells. At day 6, 2D- and 3D-cultured HTM cells (NT, non-treated control) and cultures treated with 10 ng/mL TGF-β-1, 5 ng/mL TGF-β-2, or 1 ng/mL TGF-β-3 were subjected to qPCR analysis to estimate the expression of mRNA in *TIMP1–4.* All experiments were performed in triplicate each using freshly prepared 2D HTM cells and 3D HTM spheroids (n = 15–20, total 45–60) in each experimental condition. * *p* < 0.05; ** *p* < 0.01.

**Figure 7 ijms-24-04181-f007:**
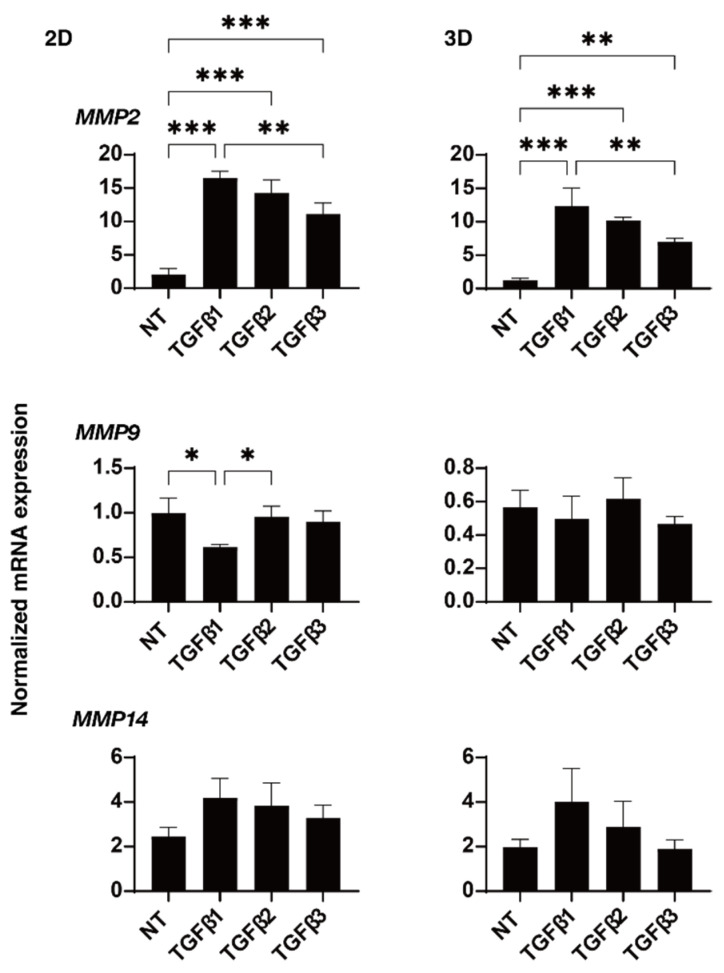
Effects of TGF-β isoforms on the mRNA expression of MMPs in 2D- and 3D-cultured HTM cells. At day 6, 2D- and 3D-cultured HTM cells (NT, non-treated control) and cultures treated with 10 ng/mL TGF-β-1, 5 ng/mL TGF-β-2, or 1 ng/mL TGF-β-3 were subjected to qPCR analysis to estimate the expression of mRNA in *MMP 2*, *9,* and *14*. All experiments were performed in triplicate each using freshly prepared 2D HTM cells and 3D HTM spheroids (n = 15–20, total 45–60) in each experimental condition. * *p* < 0.05; ** *p* < 0.01; *** *p* < 0.005.

## Data Availability

The data that support the findings of this study are available from the corresponding author upon reasonable request.

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
