# Peer review of "TGF-β-3 Induces Different Effects from TGF-β-1 and -2 on Cellular Metabolism and the Spatial Properties of the Human Trabecular Meshwork Cells"

_ijms, 2023, doi:10.3390/ijms24044181_

Round 1

Reviewer 1 Report

Overview

In this well-written pre-clinical study, Watanabe et al. investigate the role of the three known TGF-β isoforms on barrier, metabolic, and gene expression profiles 2D and 3D cultures of transformed human TM cells. The authors report that TGF-β isoforms induce, in a concentration- and isoform-dependent manner, a modest increase in TEER with relative decreases in FITC dextran permeability. By comparison, TGF-β isoforms elicited isoform-specific metabolic changes in mitochondrial respiration and glycolysis. Statistically significant, but marginal, differential effects were observed on physical properties of 3D spheroid cultures as well as mRNA expression of ECM proteins. The authors conclude that “diverse efficacies” exist among the TGF-β isoforms and that the TGF-β3 isoform may be involved in the different pathogenesis of glaucomatous TM. While the authors present an impressive amount of data using multiple experimental approaches to unmask isoform-specific responses to TGF-β signaling using 2D and 3D cultures of transformed human TM cells, the following concerns need to be addressed in order to validate these observed differences.

 Specific Comments

1.     The abstract (line 17) states “…cultures of HTM cells were compared” where HTM is defined as glaucomatous human trabecular meshwork (line 14-15). This infers that primary human TM cells from glaucoma subjects were used in this study. That is not correct. As indicated on line 338 of the Materials and Methods section, the study uses transformed human TM cells. As the use of transformed TM cells greatly limits data interpretation and meaningful clinical extrapolation, the use of transformed TM cells should be clearly stated within the abstract, and throughout, the manuscript.

2.       The manuscript would substantially benefit from the authors independent validation of commercially purchased TM cell authenticity.

3.       Extrapolating findings using transformed TM cells to the pathogenesis of glaucomatous TM (line 33) is quite a stretch and misleading. References throughout the manuscript associating experimental findings to disease relevance should be carefully considered/debated/deleted.  

4.       Figure 1, Results: These data, while interesting, are difficult to interpret and caution is advised. TEER measurements are typically obtained using primary cell cultures. The fact that transformed TM cells are exhibiting “barrier” properties is, in itself, interesting and unexpected. Might the modest differences in TEER responses observed simply reflect differences in cell density, rather than barrier integrity, in response to TGF-β stimulation?  The authors should comment on how cell density was normalized among these different experimental conditions. In addition, it is recommended that the authors re-evaluate their statistical findings for TGF-β1 results.

5.       Figure 1, Results: Permeability measurements shown in Fig. 1B are expressed as fluorescence ratio (490/530 nm) x 104. Please clarify. Moreover, there doesn’t appear to be agreement between findings shown in Fig.1A and Fig. 1B, please explain.

6.       Figure 2, Results: The use of different doses across TGF-β isoforms complicates data interpretation, particularly in context of comments made above. Concentration-dependent changes in cell responses should be considered. Alternatively, the authors would need to demonstrate isoform-specific pharmacological differences in receptor signaling to justify the comparative use of different isoform concentrations.

7.       Figure 2, Results: Cells treated with TGF-β3 exhibit an elevated OCR and display a lack of response to FCCP, suggesting that the mitochondrial respiration within these cells was aberrantly uncoupled, please explain.

8.       Figure 3: Given the variability observed (SEM shown), it is difficult to accept that statistically significant differences between isoforms occur as reported. Minor typographical error (panel B).

9.       Figures 4, 6, 7: The use of 36B4 as a reference gene should be validated under these experimental conditions.

10.   Figure 5: The representative immunohistochemical data shown in this figure should be quantified.   

11.   Collectively, the data presented throughout this manuscript argue against any sort of meaningful distinction between TGF-β1 and TGF-β2 isoforms. While it does appear that these transformed TM cells respond somewhat differently to the TGF-β3 isoform, this is most likely due to the 10-fold lower concentration used. To conclude that there are “diverse efficacies among the TGF-β isoforms” is difficult to reconcile based on the data presented. If anything, the findings of this study would seem to suggest that there are no marked difference across isoforms other than expected concentration-dependent changes.

Author Response

Dear Editor,

Thank you very much for the constructive comments concerning our manuscript, " Three TGF-β isoforms have different effects on cellular metabolism and the spatial properties of the human trabecular meshwork”. We carefully checked all of the Reviewer comments and prepared a revised version of our paper that takes these comments into account. The changes are listed below. In addition, as suggested by the Editor, English was carefully improved by a native English speaking scientist, professor Milton Feather (his certification is attached in PDF format).

Reviewer 1

Overview

In this well-written pre-clinical study, Watanabe et al. investigate the role of the three known TGF-β isoforms on barrier, metabolic, and gene expression profiles 2D and 3D cultures of transformed human TM cells. The authors report that TGF-β isoforms induce, in a concentration- and isoform-dependent manner, a modest increase in TEER with relative decreases in FITC dextran permeability. By comparison, TGF-β isoforms elicited isoform-specific metabolic changes in mitochondrial respiration and glycolysis. Statistically significant, but marginal, differential effects were observed on physical properties of 3D spheroid cultures as well as mRNA expression of ECM proteins. The authors conclude that “diverse efficacies” exist among the TGF-β isoforms and that the TGF-β3 isoform may be involved in the different pathogenesis of glaucomatous TM. While the authors present an impressive amount of data using multiple experimental approaches to unmask isoform-specific responses to TGF-β signaling using 2D and 3D cultures of transformed human TM cells, the following concerns need to be addressed in order to validate these observed differences.

 Specific Comments

  1. The abstract (line 17) states “…cultures of HTM cells were compared” where HTM is defined as glaucomatous human trabecular meshwork (line 14-15). This infers that primary human TM cells from glaucoma subjects were used in this study. That is not correct. As indicated on line 338 of the Materials and Methods section, the study uses transformed human TM cells. As the use of transformed TM cells greatly limits data interpretation and meaningful clinical extrapolation, the use of transformed TM cells should be clearly stated within the abstract, and throughout, the manuscript.

Answer; Thank you for this comment. We apologize for this confusing sentence using glaucomatous human trabecular meshwork (HTM) (line 14-15) because of my poor English. I had intended to say that our study purpose was to elucidate the pathological aspects of the TGF-b isoforms on the glaucoma etiology, especially glaucomatous HTM using normal HTM cells, in general, but not simple effects of TGF-b isoforms toward the glaucomatous HTM cells. However, since, as suggested, the current sentences may cause mis-reading, the corresponding sentence was changed; “To compare the effects among three TGF-b isoforms (TGF-b-1, TGF-b-2, and TGF-b-3) on the human trabecular meshwork (HTM), two-dimensional (2D) and three-dimensional (3D) cultures of commercially available certified immortalized HTM cells were used and the following analyses were conducted;”. In addition, as suggested, the term “commercially available certified immortalized HTM cells” was used throughout the manuscript.

  1. The manuscript would substantially benefit from the authors independent validation of commercially purchased TM cell authenticity.

Answer; Thank you for this comment. As suggested, an independent validation of the authenticity of the commercially purchased TM cells was included in the Method; “All current experiments using human tissue/cells were conducted in compliance with the tenets of the Declaration of Helsinki and were approved by the internal review board of Sapporo Medical University. Commercially available certified immortalized HTM cells (Applied Biological Materials Inc., Richmond Canada) were used in this study. The authenticity of these HTM cells were independently verified as describe in Supplemental Fig. 1.”, and new supplemental Fig. 1.

  1. Extrapolating findings using transformed TM cells to the pathogenesis of glaucomatous TM (line 33) is quite a stretch and misleading. References throughout the manuscript associating experimental findings to disease relevance should be carefully considered/debated/deleted.

Answer; Thank you for this comment. We completely agree that this sentence may cause mis-reading as the comment #1. Therefore, this was changed; “The findings presented herein suggest that these diverse efficacies among the TGF-β isoforms, especially the unique action of TGF-β-3 toward HTM may induce different effects within the pathogenesis of glaucoma.”.

  1. Figure 1, Results: These data, while interesting, are difficult to interpret and caution is advised. TEER measurements are typically obtained using primary cell cultures. The fact that transformed TM cells are exhibiting “barrier” properties is, in itself, interesting and unexpected. Might the modest differences in TEER responses observed simply reflect differences in cell density, rather than barrier integrity, in response to TGF-β stimulation? The authors should comment on how cell density was normalized among these different experimental conditions. In addition, it is recommended that the authors re-evaluate their statistical findings for TGF-β1 results.

Answer; Thank you for this comment. In terms of the normalization of the cell densities among different experimental conditions, to avoid starting to grow the immortalized HTM cells in several layers once a monolayer is reached as pointed out, TGF-b isoforms were added to the cells at 90 % confluency and thereafter cultured for 6 days, at which point, the cells reached 100 % confluent, subjected to TEER and FITC permeability measurements. Those information is included in the Method; “In terms of the normalization for cell densities among different experimental conditions, we confirmed that the numbers of the nuclear staining by DAPI were nearly comparable among these during their immunocyte chemistry.”. In addition, the nuclear staining by DAPI was counted during immunocytochemistry as shown in Fig 5. As recommended, statistical data especially FITC permeability experiment was re-evaluated as also suggested by next comment.

  1. Figure 1, Results: Permeability measurements shown in Fig. 1B are expressed as fluorescence ratio (490/530 nm) x 104. Please clarify. Moreover, there doesn’t appear to be agreement between findings shown in Fig.1A and Fig. 1B, please explain.

Answer; Thank you for this comment. As pointed out, we made a careless mistake to use ratio, but fluorescein intensity. Therefore, those were corrected. In addition, we carefully checked again both the data and replotted these data.

  1. Figure 2, Results: The use of different doses across TGF-β isoforms complicates data interpretation, particularly in context of comments made above. Concentration-dependent changes in cell responses should be considered. Alternatively, the authors would need to demonstrate isoform-specific pharmacological differences in receptor signaling to justify the comparative use of different isoform concentrations.

Answer; Thank you for this comment. In terms of this issue, we insufficiently explained why we used fixed concentrations of the TGF-b isoforms. Evidently, as shown in Fig. 1, the three TGF-b isoforms induced similar effects toward TEER measurements, but these efficacies were different, depending on the isoform. Therefore, these observations suggest that other biological functions may also be similar in the case where the concentrations of TGF-b isoforms were adjusted to induce comparable effects toward TEER measurements. Therefore, to show our study purpose more clearly, the 1st paragraph of Results was changed; “To elucidate pathological contributions of the TGF-b isoforms within glaucomatous TM, the biological effects among the TGF-b-1 ~ 3 isoforms toward 2D and 3D cell cultures of HTM cells prepared as described in our recent studies [30,32,33,38] were compared. The barrier function of the 2D cultured HTM monolayers was initially studied by measuring transendothelial electron resistance (TEER)/ FITC dextran permeability. As shown in Fig. 1, all three TGF-β isoforms had similar effects toward these analyses. These effects included significantly increased TEER values (panel A) and relatively decreased FITC dextran permeabilities (panel B) in a concentration dependent manner. However, these efficacies were different depending on the specific isoform, and the enhancement effects with respect to TEER values were nearly comparable for 10 ng/ml solutions of TGF-b-1, 5 ng/ml solutions of TGF-b-2 and 1 ng/ml solutions of TGF-b-3 (Fig. 1). These results prompted us to investigate an additional issue of whether or not other biological functions are also comparable at these TGF-β isoform concentrations. Therefore, additional experiments were conducted at these fixed TGF-β isoform concentrations.”. In terms of the cellular metabolic analysis, we agree that it is essential to evaluate several cellular functions under different concentrations of different TGF-β isoforms. Indeed, we have already assessed that the indices for cell permeability are different under different concentrations for each TGF-β isoform as shown in Figure 1. Interestingly, TGF-β3, despite being the “least” concentrated, dramatically altered cellular metabolism compared with TGF-β1 and TGF-β2 (Figure 2). This result clearly suggests that TGF-β3 induces different intracellular signaling from TGF-β1 and TGF-β2 to alter cellular metabolism. To the best of our knowledge, there have been no reports in which isoform-specific pharmacological differences in receptor signaling in HTM cells have been examined. However, a previous report by Hall et al. (J Biol Chem. 2013 Nov 1; 288(44): 32074-32092. PMID: 24056369) showed that mice lacking TGF-β1 but instead knocking in TGF-β3 did not show embryonic lethality as seen in TGF-β1-deficient mice, but had different phenotypes (shorter life span, tooth and bone loss, and altered white fat cellular metabolism), suggesting that there is at least some incompatibility between TGF-β1- and TGF-β3-induced cellular signaling. These discussions have been added in the Discussion; “In our real time cellular metabolic measurements using a Seahorse Bioanalyzer, quite interestingly, TGF-β-3, despite being the “least” concentrated, dramatically altered cellular metabolism compared with TGF-β-1 and TGF-β-2 (Figure 2). This result strongly suggests that TGF-β-3 induces a different type of intracellular signaling from TGF-β-1 and TGF-β-2 in altering cellular metabolism. To the best of our knowledge, there have been no reports in which isoform-specific pharmacological differences in receptor signaling have been examined in HTM cells. However, a previous report by Hall et al. [53] showed that mice lacking TGF-β-1 but instead knocking in TGF-β-3 did not show embryonic lethality as seen in TGF--β1-deficient mice, but had different phenotypes (shorter life span, tooth and bone loss, and altered white fat cellular metabolism), suggesting that there is at least some incompatibility between TGF-β-1- and TGF-β-3-induced cellular signaling.”. 

  1. Figure 2, Results: Cells treated with TGF-β3 exhibit an elevated OCR and display a lack of response to FCCP, suggesting that the mitochondrial respiration within these cells was aberrantly uncoupled, please explain.

Answer; We wish to express our thanks for this excellent suggestion to improve the quality of our paper. As the reviewer pointed out, treatment with TGF-β3 in HTM cells showed decreased ATP-linked respiration and increased proton leakage, with a concomitant suppression of glycolysis, suggesting that TGF-β3 induced aberrantly enhanced uncoupled respiration. Since TGF-β is known to cause an increase in the production of reactive oxygen species (ROS) via activated NADPH oxidases (NOXs) (Int J Mol Sci. 2021 Dec; 22(24): 13181. PMID: 34947978), it is possible that ROS-induced mitochondrial inner membrane damage result in aberrantly increased uncoupled respiration in TGF-β3 treated HTM cells. Another possibility is that this enhanced uncoupled respiration could have resulted from compensatory activated uncoupling proteins (UCPs) to cancel or regulate increased ROS production (Trends Endocrinol Metab. 2012 Sep;23(9):451-458. PMID: 22591987). These discussions have been added in the Discussion; “Furthermore, HTM cells that had been treated with TGF-β-3 showed decreased ATP-linked respiration and increased proton leakage, with a concomitant suppression of glycolysis, suggesting that TGF-β-3 induced aberrantly enhanced uncoupled respiration. Since it is known that TGF-β causes an increase in the production of reactive oxygen species (ROS) via activated NADPH oxidases (NOXs) [54], it is possible that ROS-induced mitochondrial inner membrane damage could result in aberrantly increased uncoupled respiration in HTM cells that were treated with TGF-β-3. Another possibility is that this enhanced uncoupled respiration could have resulted from compensatory activated uncoupling proteins (UCPs) that can cancel or regulate increased ROS production [55].”

  1. Figure 3: Given the variability observed (SEM shown), it is difficult to accept that statistically significant differences between isoforms occur as reported. Minor typographical error (panel B).

Answer; Thank you for this comment. As suggested, statistical analyses were retried and fixed. The typographic error (forse) was corrected.

  1. Figures 4, 6, 7: The use of 36B4 as a reference gene should be validated under these experimental conditions.

Answer; Thank you for this comment. As suggested, this information is included in the methods section; “Total RNA extraction followed by reverse transcription and real-time PCR were conducted were described previously [29] using specific primers and Taqman probes shown in Supplemental Table 1. For the quantification, 36B4 as a reference gene was used.

 (Information concerning the methods used in this study are shown in Supplemental Methods).”

  1. Figure 5: The representative immunohistochemical data shown in this figure should be quantified.

Answer; Thank you for this comment. As suggested, the representative immunohistochemical data were quantified and those were included in the result; “Although the statistical significance was not detected, the immunolabeling intensities of those ECM proteins were also similarly fluctuated (Fig. 5).”.

  1. Collectively, the data presented throughout this manuscript argue against any sort of meaningful distinction between TGF-β1 and TGF-β2 isoforms. While it does appear that these transformed TM cells respond somewhat differently to the TGF-β3 isoform, this is most likely due to the 10-fold lower concentration used. To conclude that there are “diverse efficacies among the TGF-β isoforms” is difficult to reconcile based on the data presented. If anything, the findings of this study would seem to suggest that there are no marked difference across isoforms other than expected concentration-dependent changes.

Answer; Thank you for this critical suggestion for improving this study. We agree that conclusion that there are “diverse efficacies among the TGF-β isoforms” is an overstatement and thus, this may lead to misunderstandings for some readers. In addition, as pointed out, among the three isoforms, the TGF-β1 and TGF-β2 isoforms showed quite similar characteristics, although these transformed TM cells respond somewhat differently to the TGF-β3 isoform even though at much lower concentrations as others. Therefore, these observations were more carefully described in the 1st paragraph of Result; “As shown in Fig. 1, all three TGF-β isoforms had similar effects toward these analyses. These effects included significantly increased TEER values (panel A) and relatively decreased FITC dextran permeabilities (panel B) in a concentration dependent manner. However, these efficacies were different depending on the specific isoform, and the enhancement effects with respect to TEER values were nearly comparable for 10 ng/ml solutions of TGF-b-1, 5 ng/ml solutions of TGF-b-2 and 1 ng/ml solutions of TGF-b-3 (Fig. 1). These results prompted us to investigate an additional issue of whether or not other biological functions are also comparable at these TGF-β isoform concentrations. Therefore, additional experiments were conducted at these fixed TGF-β isoform concentrations.”, and 2nd paragraph of Discussion; “Therefore, these collective observations rationally suggest that the TGF-b-1 ~ TGF-b-3 isoforms induce different effects toward HTM which is the one of most important biological segments that regulate AH out flow among several types of glaucoma. In the current study, to elucidate pathological effects TGF-b isoforms toward HTM, we recently developed in vitro models that replicate the structures of monolayers and multiple layers of HTM using 2D and 3D cell cultures of immortalized HTM cells. As a result, the following findings were obtained; 1) all three TGF-b isoforms increase the barrier functions of 2D HTM cell monolayers based on TEER measurements, although the efficacy of TGF-b-3 was the most potent among them. In addition, regarding the specific concentrations of each isoform required to induce the same efficacies toward TEER measurements, 2) the TGF-b-3 induced distinct effects on the cellular metabolic states of the 2D cultured HTM cells, and 3) the size or stiffness of the 3D HTM spheroids was significantly decreased in the case of TGF-b-1 or TGF-b-3, respectively, as compared with the others. Thus, these characteristic biological features of TGF-b isoforms may contribute to different types of the glaucoma pathogenesis.”.

Reviewer 2

The article by Watanabe et al looked at how the three different TGFb isoforms affected the extracelluar matrix, metabolic functions and physical properties of 2D cell cultures and 3D spheroids of cells.  While there has been a lack of characterization of all these TGFb isoforms in HTM cells, the authors used an immortalized cell strain that may not completely reflect the properties of normal HTM cells.  In addition, the discussion does not address the results and needs to be re-written.  The paper needs major revisions before it can be published.

  1. In the abstract line 18 the authors begin to use a numbering system to separate the different types of assays they used in the paper but do not continue beyond #1. Please fix this.

Answer; Thank you for this comment. As pointed, this numbering was revised; “1) transendothelial electrical resistance (TEER) and FITC dextran permeability measurements (2D), 2) a real-time cellular metabolic analysis (2D), 3) analysis of the physical property of the 3D HTM spheroids, and 4) an assessment of the gene expression levels of extracellular matrix (ECM) components (2D and 3D).”.

  1. The introduction is hard to follow and the sentences are awkward. There are too many clauses and long sentences.  This is especially true of the sentence on starting on line 42 “In terms of a possible mechanism…”  The next sentence is just as difficult to decipher starting on line 47 “Previous studies demonstrated that,…”

Answer; Thank you for this comment. As suggested, these sentences have now been revised; “The elevated levels of IOPs caused by an increase in the mechanical resistance of TM due to the deposition of excessive levels of extracellular matrix (ECM) proteins are basically associated with primary open angle glaucoma (POAG), steroid-induced glaucoma (SG) and pseudoexfoliation syndrome (PXF) [6]. Previous studies demonstrated that three different transforming growth factor-beta isoforms (TGF-β-1 ~ β-3), which are known as profibrotic cytokines, induce the formation of these excess deposits of ECM proteins in the TM [7-11].”.

  1. In the introduction lines 51-53 the authors say that TGFb2 is elevated in aqueous humor in response to glucocorticoids and cites references 12-15, which do not support this statement. The sentence continues saying this is supported by animal models and TM cell cultures, which the references 12-15 do cover.  The authors need to cite references to support that glucocorticoids increase TGFb2 in aqueous humor.  It might help to separate this sentence into two sentences to better convey what the authors intend to say.

Answer; Thank you for this critical comment. I misunderstood that “glucocorticoids increase TGFb2 in aqueous humor” was suggested but not confirmed”. Therefore this sentence was changed; “In fact, based on several studies using animal models as well as TM cell cultures, it has been suggested that elevated levels of TGF-β-2 in AH are produced in response to treatment with glucocorticoids [12-15].”.

  1. The sentence in lines 57-61 is also confusing because how it was written. I don’t really understand exactly what the authors are trying to say.  Please re-write this sentence.  It seems the authors are trying to link the effect of TGFb2 activation to PXF without referencing papers that link the two?

Answer; Thank you for this comment and we apologize for this confusing sentence. Therefore, this was changed; “In addition, recent studies have also indicated that TGF-β-1 plays an important role in the pathogenesis of PXF [19-22] which is linked with oxidative stress [23,24], ER stress response [25] and dysregulated retinoic acid signaling [26].”.

  1. The authors used immortalized TM cells in this paper. It is critical that the authors are upfront with this because immortalized TM cells do not always behave the same as primary HTM cells.  The last paragraph in the introduction the authors should mention their use.  The authors should also address this in their discussion by listing the caveats of using immortalized HTM cells.

Answer; Thank you for this comment. As suggested, we completely agree with this comment and therefore the last sentence in the introduction was changed; “Therefore, in the current study, to elucidate pathophysiological roles of three TGF isoforms (TGF-β-1, TGF-β-2, and TGF-β-3) toward human TM, we used our recently developed in vitro models using two-dimensional (2D) and three-dimensional (3D) cultures using commercially available certified immortalized HTM cells which mimic HTM monolayers and multiple sheet structures, respectively [29-33]”. In addition, as suggested, the caveats of using immortalized HTM cells was included in the last sentence in the Discussion; “the caveats of using immortalized HTM cells”.

  1. Do the immortalized TM cells used in this paper exhibit contact inhibition? Most immortalized cells will start growing in several layers once a monolayer is reached, which is not how normal HTM cells grow.  In the methods you indicate the cells are grown to 90% confluency.  Is that when the experiments were done?  At 90% confluency rather than a full monolayer?

Answer; Thank you for this comment. In terms of the timing of the TEER measurements, to avoid starting to grow the immortalized HTM cells in several layers once a monolayer is reached as pointed out, TGF-b isoforms were added to the cells at 90 % confluency and thereafter cultured for 6 days, at which point, the cells reached 100 % confluent, subjected to TEER and FITC permeability measurements.

  1. In the results section, you mention that the FITC dextran permeabilities in Figure 1B showed a concentration dependent decrease with the TGFb isoforms, but that’s not what the graphs show. It looks like all the isoforms decreased the permeabilities to the same extent regardless of what concentration was used for each isoform.  Please comment on this and change the text as needed.

Answer; Thank you for this comment. In terms of the FITC dextran revised appropriately, the data were carefully checked.

  1. Immortalized TM cells in general have characteristics that are different from their parental cell lines. For instance, they often express less amounts of extracellular matrix proteins and have a different metabolism from primary cells.  I really don’t see how studying the metabolism of an immortalized TM cell has any relevance to how a primary TM cell would behave.  While immortalized cells are a good approach to work out techniques the experiments really need to be validated in primary HTM cells.  Please repeat the metabolic studies with a primary HTM cell strain to confirm the results.

Answer; Thank you for this critical comment. We agree that the immortalized HTM cells used in this study should be different in nature from primary cultured HTM cells, including cellular metabolic states. Therefore, we also agree that it should be repeated using primary cultured HTM cells to confirm our current results. However, as my understanding, primary culture of HTM cells should be different in their nature among individual origins. In addition, since immortalized HTM cells still have characteristic biological aspects as HTM even though some differences were present as compared with the primary cultured HTM cells, I assume that the use of immortalized HTM cells still could be possible as the initial study to compare three TGF-b isoforms toward HTM cells. Therefore, as suggested, we plan to repeated the experiments using primary cultures of HTM cells obtained from several individuals with or without glaucoma in our future project. Thus, this information is included in the discussion; “However, since these concepts remain speculative at present, and currently used immortalized TM cells do not always behave the same as primary HTM cells, additional investigations using additional methodology such as RNA sequencing as well as the inhibition of specific candidate molecules by SiRNA and others, and primary HTM cells obtained from several glaucoma and non-glaucoma individuals will be required.”.

  1. Line 123 the #36 reference is incorrect. Please check which one(s) it should be.

Answer; Thank you for this comment. This was our careless mistake. Thus, this was corrected.

  1. Figure 5 has no labels what the different colors are. I am assuming the ECM protein is in green, phalloidin in red and dapi/nucleus is in blue?  The 2D images do not seem to be in focus.  Also, some of the 2D images do not look like monolayers so the ECM would not be fully formed.  Perhaps there are better images you could use?  In particular, Col1 NT, Col4 NT, Col6 TGFb1, Col6, TFb2, FN TGFb1 and FN TGFb3.  Does TGFb3 decrease phalloidin labeling?  Also, the phalloidin labeling overpowers the ECM labeling so it is difficult to see what ECM is labeled in green.  Perhaps separate the images?  In addition, there are no comments about the IF labeling in Figure 5 in the results sections.  The authors need to comment on the labeling.  The authors should also repeat these experiments using primary HTM cells because immortalized TM cells often do not form as fully mature matrix as primary HTM cells.

Answer; Thank you for these comments. As pointed out, we agree that the quality of Fig. 5 could be improved. Therefore, those figures were improved. In addition, we also agree that it will be good to add additional data using primary cultured HTM cells. However, as my understanding, the nature of primary culture of HTM cells would be different amomg individual origins. In addition, since immortalized HTM cells still have characteristic biological aspects as HTM even though some difference were present as compared with the primary cultured HTM cells, we assumed that the use of immortalized HTM cells would still be possible as the initial study to compare three TGF-b isoforms toward HTM cells. Therefore, as suggested, we would like to repeat experiments using primary cultures of HTM cell obtained from several individual with or without glaucoma in our future project. Thus, this information is included in the discussion; “However, since these concepts remain speculative at present, and currently used immortalized TM cells do not always behave the same as primary HTM cells, additional investigations using additional methodology such as RNA sequencing as well as the inhibition of specific candidate molecules by SiRNA and others, and primary HTM cells obtained from several glaucoma and non-glaucoma individuals will be required.”.

  1. The discussion is essentially a literature review and this is not the point of a discussion. The authors need to completely re-write the discussion to focus on their results and how they pertain to which type of glaucoma and relevance of their results.  What is the significance of the changes the metabolic functions?  How do changes in ECM proteins seen here relate to glaucoma?

Answer; Thank you for this critical comment. As suggested, the Discussion was rewritten to allow more focus on currently obtained data.

  1. How does a 3D spheroid of immortalized HTM cells pertain to glaucoma? Can the authors comment on how the 3D spheroid structure they are studying mimics the HTM cell environment in vivo and why it’s a good model to use?

Answer; Thank you for this comment. In terms of how the 3D spheroid structure they are studying mimics the HTM cell environment in vivo and why it’s a good model to use?, we already investigated this issue very carefully and discussed this in our recent paper (Establishment of appropriate glaucoma models using dexamethasone or TGFβ2 treated three-dimension (3D) cultured human trabecular meshwork (HTM) cells. Sci Rep 2021: 11, 19369). That is, as the maturation of the 3D HTM spheroids advanced during the 6-day culture, the normally observed down-sizing effects were further enhanced by DEX and TGFβ2, in which the effects of the latter were more evident. In addition, both drugs caused the formation of substantial ECM deposits. Concerning the Structural aspects, the 3D spheroid consists of multiple layers of HTM cells that are arranged concentrically within the 3D HTM spheroid. Based upon these results, we concluded that our established 3D HTM spheroid may replicate the multiple sheet structure of the human TM structure. In addition, the different increases in the ECM deposits by TGF-b-2 or dexamethasone within the 3D HTM spheroid suggest that those may become POAG or a type of steroid induced glaucoma model. This information is now included in the 1st paragraph of the Discussion; “As a rationale for using the 3D HTM spheroid model in addition to the conventional 2D HTM cells in the current investigation, in our recent study [30], we found that the normally observed down-sizing effects during the 6-day culture were further enhanced by dexamethasone (DEX) and TGF-β-2, and that both drugs caused the formation of substantial ECM deposits, in which these effects were more evident by TGF-β-2. Concerning structural aspects, the 3D spheroid consists of multiple layers of HTM cells that were arranged concentrically within the 3D HTM spheroid. Based upon these results, we concluded that our established 3D HTM spheroids may more accurately replicate multiple sheet structure of the human TM structure. In addition, the different increase of ECM deposits by TGF-b-2 or DEX within the 3D HTM spheroid suggests that these may become POAG or a steroid induced glaucoma model.”.

  1. In the supplemental methods the authors repeat the “analysis of real-time cellular metabolism of the 2D-cultured HTM cells by a Seahorse Bioanalyzer. Please remove the duplicate.

Answer; Thank you for this comment. As pointed out, the duplicate of this methodology within the supplemental materials was removed.

Reviewer 2 Report

The article by Watanabe et al looked at how the three different TGFb isoforms affected the extracelluar matrix, metabolic functions and physical properties of 2D cell cultures and 3D spheroids of cells.  While there has been a lack of characterization of all these TGFb isoforms in HTM cells, the authors used an immortalized cell strain that may not completely reflect the properties of normal HTM cells.  In addition, the discussion does not address the results and needs to be re-written.  The paper needs major revisions before it can be published.

1.       In the abstract line 18 the authors begin to use a numbering system to separate the different types of assays they used in the paper but do not continue beyond #1.  Please fix this.

2.       The introduction is hard to follow and the sentences are awkward.  There are too many clauses and long sentences.  This is especially true of the sentence on starting on line 42 “In terms of a possible mechanism…”  The next sentence is just as difficult to decipher starting on line 47 “Previous studies demonstrated that,…”

3.       In the introduction lines 51-53 the authors say that TGFb2 is elevated in aqueous humor in response to glucocorticoids and cites references 12-15, which do not support this statement.  The sentence continues saying this is supported by animal models and TM cell cultures, which the references 12-15 do cover.  The authors need to cite references to support that glucocorticoids increase TGFb2 in aqueous humor.  It might help to separate this sentence into two sentences to better convey what the authors intend to say.

4.       The sentence in lines 57-61 is also confusing because how it was written.  I don’t really understand exactly what the authors are trying to say.  Please re-write this sentence.  It seems the authors are trying to link the effect of TGFb2 activation to PXF without referencing papers that link the two?

5.       The authors used immortalized TM cells in this paper.  It is critical that the authors are upfront with this because immortalized TM cells do not always behave the same as primary HTM cells.  The last paragraph in the introduction the authors should mention their use.  The authors should also address this in their discussion by listing the caveats of using immortalized HTM cells.

6.       Do the immortalized TM cells used in this paper exhibit contact inhibition?  Most immortalized cells will start growing in several layers once a monolayer is reached, which is not how normal HTM cells grow.  In the methods you indicate the cells are grown to 90% confluency.  Is that when the experiments were done?  At 90% confluency rather than a full monolayer?

7.       In the results section, you mention that the FITC dextran permeabilities in Figure 1B showed a concentration dependent decrease with the TGFb isoforms, but that’s not what the graphs show.  It looks like all the isoforms decreased the permeabilities to the same extent regardless of what concentration was used for each isoform.  Please comment on this and change the text as needed.

8.       Immortalized TM cells in general have characteristics that are different from their parental cell lines.  For instance, they often express less amounts of extracellular matrix proteins and have a different metabolism from primary cells.  I really don’t see how studying the metabolism of an immortalized TM cell has any relevance to how a primary TM cell would behave.  While immortalized cells are a good approach to work out techniques the experiments really need to be validated in primary HTM cells.  Please repeat the metabolic studies with a primary HTM cell strain to confirm the results. 

9.       Line 123 the #36 reference is incorrect.  Please check which one(s) it should be.

10.   Figure 5 has no labels what the different colors are.  I am assuming the ECM protein is in green, phalloidin in red and dapi/nucleus is in blue?  The 2D images do not seem to be in focus.  Also, some of the 2D images do not look like monolayers so the ECM would not be fully formed.  Perhaps there are better images you could use?  In particular, Col1 NT, Col4 NT, Col6 TGFb1, Col6, TFb2, FN TGFb1 and FN TGFb3.  Does TGFb3 decrease phalloidin labeling?  Also, the phalloidin labeling overpowers the ECM labeling so it is difficult to see what ECM is labeled in green.  Perhaps separate the images?  In addition, there are no comments about the IF labeling in Figure 5 in the results sections.  The authors need to comment on the labeling.  The authors should also repeat these experiments using primary HTM cells because immortalized TM cells often do not form as fully mature matrix as primary HTM cells. 

11.   The discussion is essentially a literature review.  That’s not the point of a discussion.  The authors need to completely re-write the discussion to focus on their results and how they pertain to which type of glaucoma and relevance of their results.  What is the significance of the changes the metabolic functions?  How do changes in ECM proteins seen here relate to glaucoma?

12.   How does a 3D spheroid of immortalized HTM cells pertain to glaucoma?  Can the authors comment on how the 3D spheroid structure they are studying mimics the HTM cell environment in vivo and why it’s a good model to use?

13.   In the supplemental methods the authors repeat the “analysis of real-time ellular metabolism of the 2D-cultured HTM cells by a Seahorse Bioanalyzer.  Please remove the duplicate.

Author Response

(The authors gave the same response as above.)

Round 2

Reviewer 1 Report

While the authors have responded to most of the concerns raised, and the revised manuscript is much improved, I still would ask that the authors respectfully address a few remaining concerns:

1. Figure 1: Other than TGFb3, I don't understand how the authors claim pharmacologic dose dependency. Clearly, from the data presented, TGFb isoforms 1 & 2 do not exhibit dose dependency. At 1ng/ml, TGFb2 response was indistinguishable from background control. At 5 ng/ml, it was maximal. This does not constitute a pharmacological dose-response study. I think they need to re-state this observation to be more in line with what the data are showing.

2. Figure 1: I still have concerns regarding normalization. No information is given as to the number of cells plated and no quantitative data is offered that would demonstrate normalization across conditions. These minor changes in TEER / Permeability may just be a difference in cell density. It would be helpful to see cell proliferation data (MTT assay) under these same experimental condition to verify that what is being observed is not due to changes in cell density of these transformed cells.

3. Figure 2: For the authors to claim differential responses with TGFb3 isoform, they need to justify why cells that received TGFb3 isoform treatment were uncoupled even before treatment began (time 0). This suggests that these cells were somehow compromised before treating them with TGFb3, thus limiting data interpretation.

4. I would strongly recommend that the title of the manuscript be revised to reflect the data presented including (i) the use of transformed TM cells and (ii) that only TGFb3 isoform may have slightly different responses from that of TGFb1 and TGFb2.

5. Lastly, all of the observations presented in this manuscript could easily be explained by a difference in isoform concentration. I don't believe the authors have convincingly demonstrated a pharmacological dose-response uniqueness among these isoforms. If, at the same concentration (10 ng/ml), there are distinct differences in outcome responses across isoforms, then I should think the manuscript would be more amenable to critical review.

Author Response

Dear Editor,

Thank you very much for the constructive comments concerning our manuscript, " Three TGF-β isoforms have different effects on cellular metabolism and the spatial properties of the human trabecular meshwork”. We carefully checked all of the Reviewer comments and prepared a revised version of our paper that takes these comments into account. The changes are listed below.

Reviewer 1

While the authors have responded to most of the concerns raised, and the revised manuscript is much improved, I still would ask that the authors respectfully address a few remaining concerns:

  1. Figure 1: Other than TGFb3, I don't understand how the authors claim pharmacologic dose dependency. Clearly, from the data presented, TGFb isoforms 1 & 2 do not exhibit dose dependency. At 1ng/ml, TGFb2 response was indistinguishable from background control. At 5 ng/ml, it was maximal. This does not constitute a pharmacological dose-response study. I think they need to re-state this observation to be more in line with what the data are showing.

Answer; Thank you for this critical comment. We agree with this comment and therefore, we revised the section regarding this issue to make it more understandable and more precise, as follows; “As shown in Fig. 1, the effects of TGF-b-3 included significantly increased TEER values (panel A) and relatively decreased FITC dextran permeabilities (panel B) and these effects were concentration-dependent. However, in contrast to TGF-b-3, TGF-b-1 and TGF-b-2 exhibited similar but distinctly different effects within a concentration range of 1-10 ng/ml. That is, 1) the TGF-b-1 induced increase in the TEER values was higher at a concentration of 10 ng/ml than at the 1 ng/ml, and 2) the TGF-b-2 induced increase in the TEER values reached a plateau at a concentration of 5 ng/ml but no increasing effects were detected at a concentration of 1 ng/ml. Alternatively, the FITC-dextran permeabilities fluctuated in an opposite manner to the TEER values despite the fact that the differences were not significant. Taken together, although these effects were different depending on the specific isoform, the enhancement effects with respect to TEER values were nearly comparable for 10 ng/ml solutions of TGF-b-1, 5 ng/ml solutions of TGF-b-2 and 1 ng/ml solutions of TGF-b-3 (Fig. 1). This provided us with a rationale for comparing other biological functions among the three TGF-b isoforms. Therefore, additional experiments were conducted at these fixed TGF-β isoform concentrations (10 ng/ml TGF-b-1, 5 ng/ml TGF-b-2 and 1 ng/ml TGF-b-3).”.

  1. Figure 1: I still have concerns regarding normalization. No information is given as to the number of cells plated and no quantitative data is offered that would demonstrate normalization across conditions. These minor changes in TEER / Permeability may just be a difference in cell density. It would be helpful to see cell proliferation data (MTT assay) under these same experimental condition to verify that what is being observed is not due to changes in cell density of these transformed cells.

Answer; Thank you for this comment. As suggested, this information related to counting numbers of cell densities stained by DAPI is now included; In the presence or absence of 1, 5 or 10 ng/ml TGF-β-1, TGF-β-2 or TGF-β-3, the 2D cultured HTM monolayers were subjected to TEER and FITC permeability measurements. Plots of the TEER values (Ωcm2) are shown in panel A, and the fluorescein intensities at an excitation wavelength of 490 and an emission wavelength of 530 nm are plotted in panel B. The numbers of cells within the 100 mm x 100 ml squares (n= 10 different area) in the TEER 2D planar culture plate (NT; 4.2±1.17, TGF-b-1; 4.4±1.85, TGF-b-2; 5.0±1.53, TGF-b-3; 4.6±0.93). All experiments were performed in triplicate using fresh preparations (n=4). * P<0.05, ** P<0.01, *** P<0.005.”.

  1. Figure 2: For the authors to claim differential responses with TGFb3 isoform, they need to justify why cells that received TGFb3 isoform treatment were uncoupled even before treatment began (time 0). This suggests that these cells were somehow compromised before treating them with TGFb3, thus limiting data interpretation.

Answer; Thank you for this comment. In Fig. 2, the "time 0" refers to the time when a Seahorse Bioanalyzer started measuring oxygen consumption, but not the time of exposure to TGF-β. Our real-time metabolic analysis shown in Fig. 2 was performed in the same assay buffer without TGF-β on 2D-cultured HTM cells that were previously treated with different TGF-β isoforms for 6 days. Since the expression of the Method section in the previous version may have led to misleading text, this has now been revised; “The real-time cellular metabolic function analysis involved collecting data on the oxygen consumption rate (OCR) and extracellular acidification rate (ECAR) of the 2D HTM cells that had been cultured in the absence and presence of the three TGF-β isoforms for 6 days using a Sea-horse XFe96 Bioanalyzer (Agilent Technologies, Santa Clara, CA, U.S.A.) as described recently [35-37]. (Information concerning the methods used in this study are shown in Supplemental Methods).”.

  1. I would strongly recommend that the title of the manuscript be revised to reflect the data presented including (i) the use of transformed TM cells and (ii) that only TGFb3 isoform may have slightly different responses from that of TGFb1 and TGFb2.

Answer; Thank you for this comment. We agree with this comment, and therefore, title was changed; “TGF-β-3 induces different effects from TGF-β-1 and -2 on cellular metabolism and the spatial properties of the human trabecular meshwork”.

  1. Lastly, all of the observations presented in this manuscript could easily be explained by a difference in isoform concentration. I don't believe the authors have convincingly demonstrated a pharmacological dose-response uniqueness among these isoforms. If, at the same concentration (10 ng/ml), there are distinct differences in outcome responses across isoforms, then I should think the manuscript would be more amenable to critical review.

Answer; Thank you for this comment. We understand that using the same concentrations should be the standard strategy for evaluating three TGF-b isoforms as suggested. Nevertheless, as demonstrated in Fig. 1, huge differences in the TEER values were observed among three TGF-b isoforms in the case when the same concentrations of 1, 5 or 10 ng/ml. If the main study purpose is jest comparison of TEER values among them, this strategy may be suitable. However, if the main study purpose is to compare several different biological functions among them, in addition to the standard strategy using the same concentrations, we assume that our strategy to use some different concentrations to fix the standard among them in the one of the investigations, and to analyze other biological functions could alternatively allow us to compare the several biological aspects among them. However, the proposed strategy still be important advice is worthy of being investigated, and therefore, this information is included in the last paragraph of the Discussion; “However, since these concepts remain speculative at present, and currently used immortalized TM cells do not always behave the same as primary HTM cells, additional investigations using additional methodology such as RNA sequencing as well as the inhibition of specific candidate molecules by SiRNA and others, in addition to using the same concentrations of TGF-b isoforms, and primary HTM cells obtained from several glaucoma and non-glaucoma individuals will be required.”

Reviewer 2

Thank you for addressing our concerns.  The paper is much improved, however there are still a few areas that need to be addressed.

  1. Why is melanoma a keyword? Perhaps the keywords need to be updated/changed?

Answer; Thank you for this comment, and we apologize careless mistake. As suggested the key words were updated/changed; “3D spheroid cultures, 2D planar culture, human trabecular meshwork (HTM), TGF-b, real-time cellular metabolic analysis”

  1. In the introduction line 44 the wrong reference is used. The authors need to reference papers that talked about excessive ECM in POAG, SG and PXF but the reference cited has to do with MYOC expression in TM cells.

Answer; Thank you for this comment and we apologize for not citing the correct reference. As suggested, another suitable reference was cited; “Kate E Keller and Donna M Peters. Pathogenesis of glaucoma: Extracellular matrix dysfunction in the trabecular meshwork-A review. Clin. Exp. Ophthalmol. 2022 Mar;50(2):163-182. doi: 10.1111/ceo.14027.”

  1. In the introduction line 46 references 7-11. The authors cite these to support the excess deposition of ECM proteins in the TM yet most of the references don’t study TM cells.  Please update.

Answer; Thank you for this comment and we apologize for not citing suchsuitable references. As far as we survey, among three TGF-b isoforms, TGF-b-1 and TGF-b-2 induced effects toward ECM deposits within TM were reported, but TGF-b-3 induced effects have not been well characterized. Therefore, as suggested, the corresponding sentence was changed and other suitable references were cited (Robert J Wordinger, Tasneem Sharma, Abbot F Clark. The role of TGF-β2 and bone morphogenetic proteins in the trabecular meshwork and glaucoma. J Ocul Pharmacol Ther. 2014 Mar-Apr;30(2-3):154-62. doi: 10.1089/jop.2013.0220. Xin Wang, Guoli Huai, Hailian Wang, Yuande Liu, Ping Qi, Wei Shi, Jie Peng, Hongji Yang, Shaoping Deng, Yi Wang. Mutual regulation of the Hippo/Wnt/LPA/TGF‑β signaling pathways and their roles in glaucoma (Review). Int J Mol Med. 2018 Mar;41(3):1201-1212. doi: 10.3892/ijmm.2017.3352.); “Previous studies demonstrated that among three different transforming growth factor-beta isoforms (TGF-β-1 ~ β-3) known as profibrotic cytokines, TGF-β-1 and TGF-β-2 induce the formation of these excess deposits of ECM proteins in the TM[7,8].”.

  1. The images in Figure 5A are still very much out of focus. Perhaps showing higher magnifications or blowing up the images would improve the figure.  As it is now, I can’t conclude anything from the images.  I can’t see the fibrillar nature of the ECM proteins so can’t conclude if a normal ECM has been established. In fact, the fibronectin label looks intracellular.  It doesn’t appear that any of the fibronectin is in the ECM of these cell cultures.  The authors also need to state in the figure legend how the stain intensity was determined and what the graphs are relative to – no treatment?  Looking at the stain intensity graphs of the 3D cultures, it doesn’t appear the TGFb isoforms had much effect on ECM protein.  Please comment.

Answer; Thank you for this comment. As pointed out, Fig. 5A was improved by changing images with higher magnification (40 x), and to show the fibrillar nature of the ECM proteins. In addition, the methodology used to evaluate the fluorescein labeling levels was also included; “To quantify the stain intensities of each target molecule of the 2D cultured cells, 1) their stain intensities among the observed areas were evaluated using Image J (NIS-Elements 4.0 software), 2) numbers of nuclei stained by DAPI within those areas were counted, 3) the stain intensities were divided by the numbers of nuclei, and 4) the relative intensities of each experimental condition were calculated by comparing with those of the control. For 3D spheroids, serial-axis images with a 2.2 μm interval at a height of 35 μm from their surface were obtained. The maximum intensity/surface area among the above observed areas was calculated using Image J (NIS-Elements 4.0 software) as follows: surface area=D×A/(A+π×H2), where D (μm) indicates spheroid diameter, A (μm2) indicates the area of the sectioned spheroid, and H (μm) indicates the height (=35 μm). For estimating the numbers of cells within a 3D spheroid, the volume of a 3D spheroid and the volume of a representative cell was calculated by assuming a spherical shape and the tentative diameters were estimated by largest cross-section of phalloidin images of the 3D spheroid (n=5) and the distance between two adjacent nuclei stained by DAPI (n=5 for one section and was repeated five times using different preparations), respectively. The relative intensities of each experimental condition were calculated by comparing with those of the control.”. In addition, the discrepancy between stain intensities and gene expression of the 3D spheroid, possible explanation was included in the Result; “As possible reason for the discrepancy between the stain intensities and the gene expression levels of the 3D spheroid specimens, it was speculated that the immunolabeling reflected the target molecules localized within the surface of the 3D spheroid, but in contrast, the gene expressions analysis detected the levels of the target molecules within whole 3D spheroid. In fact, such discrepancy was also observed in our previous studies [41,42].”.

  1. The authors need to be more careful on reporting their results in Figures 6-7. For instance, in line 175 the authors say TIMP1 was upregulated but the graph shows very little change and it wasn’t significant.  Also, TIMP3 was only up significantly with TGFb1 in 2D cultures and not 3D cultures.  MMP9 was only downregulated by TGFb1.

Answer; Thank you for this comment. As suggested, the results related to Figs 6 and 7 were more precisely and more straightforwardly described; “Similar to this, dominant effects of TGF-b-1 or -2 were also observed in the mRNA expressions of some of the ECM modulators, TIMPs (Fig. 6) and MMPs (Fig. 7). That is, a significant up-regulation of TIMP 2 (3D) by TGF-b-1 or -2, TIMP3 (2D) by TGF-b-1, and MMP2 (2D and 3D) by TGF-b-1 or -2, and the down-regulation of MMP9 by TGF-b-1 (2D) were observed.”.

  1. Lastly, I realized the authors did not indicate which antibodies were used in Figure 5 in the supplemental methods. Also, did the authors use the same labeling procedure (i.e. times of incubations) with the 2D and 3D cultures?

Answer; Thank you for this comment. As suggested, details of the antibodies are now included, and basically same procedures were used except microscopic detection; “Immunocytochemistry of 2D cultured HTM cells and 3D HTM spheroids was examined by previously described methods, with minor modifications[38][39]. All procedures were performed at room temperature unless otherwise stated. Briefly, 2D HTM cells were cultured on glass slides (Lab-Tek II Chamber slide, Thermo Fisher Scientific Inc.) or 3D HTM spheroids prepared as described above under several experimental conditions were fixed in 4 % paraformaldehyde overnight in PBS, blocked in 3 % BSA in PBS for 3 hours, washed twice with PBS for 30 minutes. These were then reacted with an anti-human COL1 (#600-401-103-0.1, Rockland antibodies &assays, Limerick, PA U.S.A.), COL4 (#600-401-108-0.1, Rockland antibodies &assays, Limerick, PA U.S.A.), COL6 (#600-401-106-0.1, Rockland antibodies &assays, Limerick, PA U.S.A.) or FN (#sc-8422, Santa Cruz Biotechnology, INC., Dallas, TX U.S.A.) rabbit antibody (1:200 dilutions) at 4°C overnight. After washing 3 times with PBS for 1 hour each, they were then reacted with a 1:1000 dilution of a goat anti-rabbit IgG (488 nm, # A-11008, ThermoFischer Scientific, Waltham, MA U.S.A.), phalloidin (594 nm, #20553, Cyman Chemical, Ann Arbor MI U.S.A.) and DAPI (#28718-90-3, Dojindo, Kumamoto, Japan) for 3 hrs, and thereafter mounted with ProLong Gold Antifade Mountant with a cover glass. Immunofluorescent images were obtained by means of a Nikon A1 confocal microscope using a ×20 air objective with a resolution of 1024 × 1024 pixels. For 3D spheroids, serial-axis images with a 2.2 μm interval at a height of 35 μm from their surface were obtained. The maximum intensity/surface area among the above observed areas was calculated using Image J (NIS-Elements 4.0 software) as follows: surface area=D×A/(A+π×H2), where D (μm) indicates spheroid diameter, A (μm2) indicates the area of the sectioned spheroid, and H (μm) indicates the height (=35 μm). For estimating the numbers of cells within a 3D spheroid, the volume of a 3D spheroid and the volume of a representative cell was calculated by assuming a spherical shape and the tentative diameters were estimated by largest cross-section of phalloidin images of the 3D spheroid (n=5) and the distance between two adjacent nuclei stained by DAPI (n=5 for one section and was repeated five times using different preparations), respectively.”.

Reviewer 2 Report

Thank you for addressing our concerns.  The paper is much improved, however there are still a few areas that need to be addressed.

1.       Why is melanoma a keyword?  Perhaps the keywords need to be updated/changed?

2.       In the introduction line 44 the wrong reference is used.  The authors need to reference papers that talked about excessive ECM in POAG, SG and PXF but the reference cited has to do with MYOC expression in TM cells.

3.       In the introduction line 46 references 7-11.  The authors cite these to support the excess deposition of ECM proteins in the TM yet most of the references don’t study TM cells.  Please update.

4.       The images in Figure 5A are still very much out of focus.  Perhaps showing higher magnifications or blowing up the images would improve the figure.  As it is now, I can’t conclude anything from the images.  I can’t see the fibrillar nature of the ECM proteins so can’t conclude if a normal ECM has been established. In fact, the fibronectin label looks intracellular.  It doesn’t appear that any of the fibronectin is in the ECM of these cell cultures.  The authors also need to state in the figure legend how the stain intensity was determined and what the graphs are relative to – no treatment?  Looking at the stain intensity graphs of the 3D cultures, it doesn’t appear the TGFb isoforms had much effect on ECM protein.  Please comment.

5.       The authors need to be more careful on reporting their results in Figures 6-7.  For instance, in line 175 the authors say TIMP1 was upregulated but the graph shows very little change and it wasn’t significant.  Also, TIMP3 was only up significantly with TGFb1 in 2D cultures and not 3D cultures.  MMP9 was only downregulated by TGFb1.

6.       Lastly, I realized the authors did not indicate which antibodies were used in Figure 5 in the supplemental methods.  Also, did the authors use the same labeling procedure (i.e. times of incubations) with the 2D and 3D cultures?

Author Response

(The authors gave the same response as above.)

Round 3

Reviewer 1 Report

no further comments

Author Response

Dear Editor,

Thank you very much for the constructive comments concerning our manuscript, " Three TGF-β isoforms have different effects on cellular metabolism and the spatial properties of the human trabecular meshwork”. We carefully checked all of the Reviewer comments and prepared a revised version of our paper that takes these comments into account. The changes are listed below.

Reviewer 2

Thank you for your corrections.  There are a few minor issues remaining. 

  1. The title change is more in line with what the paper is reporting however please change trabecular meshwork to trabecular meshwork cells since you are not studying the tissue.

Answer; Thank you so much for this comment. As suggested, the title was changed to “TGF-β-3 induces different effects from TGF-β-1 and -2 on cellular metabolism and the spatial properties of the human trabecular meshwork cells.”.

  1. Figure 5 is better, but there is a lot of background green fluorescence showing in all the 2D images making it seem like there is a lot more labeling then there actually is. Matrix proteins should be brightest and fibrillar in nature around the outside of the cells and sometimes over the top of cells, not intracellular which is what most of the green is, especially in the fibronectin images.  I'm concerned the intensities measured are seeing background fluorescence rather than real labeling.  However, the trends in the intensity graphs seem to reflect the images shown.  Please decrease the contrast/brightness to bring down the background labeling.  Also, the Figure 5 legend does not mention the Image J results.  Please add this.

Answer; Thank you so much for this comment. As suggested, the contrast/brightness was decreased to bring down the background labeling. In addition, the missing information related to the Image J result in the Fig. 5 legend was included; “At Day 6, 2D and 3D cultured HTM cells (NT; non-treated control) and cultures treated with 10 ng/ml TGF-β-1, 5 ng/ml TGF-β-2 or 1 ng/ml TGF-β-3 were subjected to immunofluorescein labeling of ECMs (COL1, COL4, COL6, FN and aSMA). All experiments were performed in duplicate using fresh preparations (n=5, total 10). Representative merged images with DAPI and phalloidin are shown (panel A left; 2D, panel B left; 3D, scale bar; 100 mm). The immunostaining levels of each target molecule among the observed areas were evaluated using Image J (NIS-Elements 4.0 software) and plotted (panel A right; 2D, panel B right; 3D)”.  

Reviewer 2 Report

Thank you for your corrections.  There are a few minor issues remaining.  

1.  The title change is more in line with what the paper is reporting however please change trabecular meshwork to trabecular meshwork cells since you are not studying the tissue.

2.  Figure 5 is better, but there is a lot of background green fluorescence showing in all the 2D images making it seem like there is a lot more labeling then there actually is.  Matrix proteins should be brightest and fibrillar in nature around the outside of the cells and sometimes over the top of cells, not intracellular which is what most of the green is, especially in the fibronectin images.  I'm concerned the intensities measured are seeing background fluorescence rather than real labeling.  However, the trends in the intensity graphs seem to reflect the images shown.  Please decrease the contrast/brightness to bring down the background labeling.  Also, the Figure 5 legend does not mention the Image J results.  Please add this.

Author Response

(The authors gave the same response as above.)
